# Interfacing Arduino Boards with Optical Sensor Arrays: Overview and Realization of an Accurate Solar Compass

**DOI:** 10.3390/s23249787

**Published:** 2023-12-12

**Authors:** Daniele Murra, Sarah Bollanti, Paolo Di Lazzaro, Francesco Flora, Luca Mezi

**Affiliations:** ENEA, Italian National Agency for New Technologies, Energy and Sustainable Economic Development, Fusion and Technology for Nuclear Safety and Security Department, Frascati Research Center, Via E. Fermi 45, 00044 Frascati, Italy

**Keywords:** Arduino board, photosensors, array detector, solar compass

## Abstract

In this paper, an overview of the potentiality of Arduino boards is presented, together with a description of the Arduino interfacing with light multi-sensors. These sensors can be arranged in linear arrays or in a matrix configuration (CCD or CMOS type cameras) and are equipped with tens, hundreds, or even thousands of elements whose sizes range from a few microns to tens of microns. The use of these sensors requires electronics that have high time accuracy, since they work through regular pulses sent by an external source and, furthermore, have the ability to digitize and store voltage signals precisely and quickly. We show that, with the appropriate settings, a simple Arduino board can handle both 1D and 2D optical sensors. Finally, we describe a solar compass made with such a board coupled to one of the tested optical array sensors that is capable of providing the north direction with a very high degree of accuracy.

## 1. Introduction

The hardware platform called ‘Arduino’ has become a standard for use among electronics amateurs as well as in research laboratories. The reasons for its success are its ease of use, the versatility of its applications, the countless peripherals that can be connected to it, and the copious literature consisting of algorithms, libraries, forums, and help of all kinds that can be found on the Internet [1], together with scientific papers [2,3,4] and books [5,6]. 

Nowadays, many research laboratories exploit the Arduino electronics, but in most cases, it is used as an analog-to-digital converter and storage device [7], as a temperature datalogger [8], as a servo-mechanism controller [9], as a brightness and contrast regulator [10], as a triggering system [11], as an RGB sensor controller [12], and so on. In contrast, information on the interface between Arduino and an optical sensor array is difficult to find in the scientific literature. Some examples involve an apparatus including 32 light-dependent resistors [13] or the use of the 128-pixel TSL1401CL array [14,15]. 

In this paper, we briefly introduce the board hardware and some applications in a research laboratory, and then we focus on the use of the board in interfacing with 1D- and 2D-array light sensors. The optical sensors considered here are standard silicon sensors with a spectral response ranging from infrared to approximately 300 nm. However, some important advances in this field should be taken into account when an electro-optical device has to be designed, especially for UV detection [16,17].

In our laboratory, the main goal was the possibility of using a photosensor array to measure the viewing angle of a light source from a specific observation point with extreme accuracy, as in the case of the solar compass that ENEA designed and patented [18,19]. In the case of the solar compass, the light sensor was initially made up of a webcam managed by a laptop PC. Then, this was replaced, respectively, by a matrix sensor and a microcontroller communicating via a serial protocol. The time required to capture a frame to measure the viewing angle of the sun and download and analyze it is about 10 s. This fact led us to search for possible alternatives, both for the sensor and the electronics, possibly also improving other aspects of the device, such as the use of a graphic display or communicating with a mobile phone via Bluetooth. Moreover, due to a specific request to design a new compass prototype, we found that some components are no longer commercially available; therefore, we searched for other sensors and electronic boards. Our choice fell on an Arduino board and linear photosensors. 

## 2. The Arduino Board

The main features of Arduino boards are described in numerous books and articles [1,2,3,4,5,6,7,8,9,10,11,12,13,14,15]. Figure 1 shows a photo of an Arduino UNO board, where the various elements that make it up are indicated. It is possible to program the board microcontroller through the classic connection to an external programmer, as happens with a common PIC (Programmable Interrupt Controller). More simply, this can also be performed through the dedicated development environment, which, once installed on the PC, allows us to edit a C/C++ program, compile it, and send it directly to the board via the USB serial port.

Looking at the number of digital (input/output) and analogue ports (input only, except for the DUE board, where there are also analogue output ports), it is evident that applications in which it is necessary to connect many sensors or actuators do not represent a problem.

In the Laboratory of Plasma Applications and Interdisciplinary Experiments at the ENEA Research Center in Frascati, several experiments made use of Arduino boards, as in [8], where UV light irradiation was carried out on samples to test their resistance to solar radiation degradation outside the atmosphere. In that case, an Arduino NANO board was interfaced both with some light sensors (photoresistors) to monitor the intensity of the light as a function of time and with a probe to control the temperature of the sample and possibly turn off the light if the temperature had exceeded the alert value.

On another occasion, an Arduino UNO board was used to drive linear actuators to simulate the movements of a structure. This is suitable for verifying the behavior of optical fiber sensors under mechanical stress. After a positive final test, the box containing the power supplies, Arduino, and the control console was delivered to an ENEA group that used it within a national project [20] aimed at an experimental study for a seismic isolation system of structures equipped with and monitored by fiber optic sensors.

Finally, an Arduino board was used for characterization measurements of a UV-C LED sanitization system. The aim was to measure the angular distribution of the light energy emitted by an LED. For this purpose, we arranged a setup where the LED was rotating around a vertical axis lying on the LED emitting plane, while a photodiode was fixed. An Arduino NANO board was equipped with a motor driver, allowing a stepper motor to rotate the LED, while the photodiode output was connected, via a series resistor, to an analog input on the same board. The Arduino program moved the motor by a certain angle and, at the same time, measured the voltage from the photodiode, recording both the angle and the light intensity, in such a way to reconstruct the angular distribution of the LED.

The advantages of using a programmable board equipped with several interfaces and to which a large number of sensors can be connected are evident. The following paragraphs illustrate the feasibility of using these boards to also drive light sensors made up of hundreds or thousands of sensors, arranged in linear arrays or matrices. The use of these sensors is a more challenging task, because for their correct functioning, it is necessary to provide a clock signal of tens of kHz or more, and furthermore, the amount of data generated by these sensors can exceed the memory size of the processors themselves.

## 3. Light Sensors

A very simple light sensor, the photoresistor, whose electrical resistance varies as a function of the light power that excites it, was used during the experiment described in [8]. When this sensor is inserted into an electrical circuit, the light intensity is measured by checking the voltage drop across the resistor pins.

Other common light sensors are photodiodes, made up of a semiconductor junction, in which the impinging light allows, through the voltage-powered junction, the passage of current, which flows through a series-connected resistor, providing the voltage signal at its ends. Compared to the photoresistor, the operating principle is different, but from a measurement point of view, it is very similar. The situation is unlike that of the linear array or matrix sensors. In this case, there are hundreds of photodiodes aligned along a row or thousands of cells arranged in a grid. These sensors are internally equipped with an electrical circuit, which ‘downloads’ the signals coming from the individual microdetectors and sends them sequentially to the external processor. As anticipated in the Introduction, the need to use a sensor of this type arose during the development of an electronic solar compass in order to measure, with extreme precision, the arrival direction of the sun’s rays with respect to a vertical reference plane and then to determine the true north direction. Consequently, an investigation was conducted to verify which sensors were available on the market and which of these could be compatible with the Arduino hardware. 

The linear sensors taken into consideration were the following: -TSL1401CL by AMS;-LF1401 by IC HAUS;-S9226 by Hamamatsu;-ILX554A from Sony;-TCD1304DG from Toshiba.

The option of using a 2D-array was also explored, in particular, the Omnivision OV7670 CMOS (Complementary Metal-Oxide Semiconductor) camera. Table 1 shows the main characteristics of the various sensors examined. It is clear that, apart from the AMS and IC Haus sensors, which are practically identical, their characteristics vary a lot from each other, both in terms of geometry, in particular, the size of the pixels, and in terms of the clock speed for the operation of the internal circuit of the chip that drives the photodiodes. The cost of these sensors is quite low, and some of them can be found in the catalogs of large electronics vendors on the Internet. The request for a high spatial resolution leads to the choice of Sony or Toshiba chips, while the need to use a low clock time while maintaining the execution of a measurement within a few fractions of a second requires the number of pixels to not be excessively high. In the last case, the choice would hence fall on the Hamamatsu array or on the twins of AMS or IC Haus. Clearly, if the purpose of the measurement is to obtain an image or if two-dimensional information is needed, the only possibility is to opt for the CMOS camera. As we show in the next paragraph, with Arduino, it was possible to drive all of the sensors listed in Table 1, except for the Toshiba one, probably due to pulse synchronization issues. 

## 4. The Characteristics of Arduino and the Requirements of Linear Sensors

The photodiode arrays are made up of a light-sensitive area, where hundreds or thousands of sensors are aligned, and an externally controlled electronic circuit, which regulates the operation of the electro-optical elements and sends voltage data proportional to the local value of the light intensity. These data are sent sequentially, and timing synchronization occurs via a clock signal provided by the external driver. In the case of the CMOS matrix, the external clock is needed to not only synchronize the sending of the intensity data but also to allow all the operations of the on-board electronics. The number of communication pins between the driver and sensor is always three for all models, except for the IC Haus and the Toshiba models. The three pins represent the clock signal (input), start signal (input), and output signal. In the IC Haus and Toshiba models, there is an additional input signal to regulate the light integration time. The temporal graph representing the sequence of signals to be sent to the arrays is represented in Figure 2 in its most general form (the true sequence may vary slightly from sensor to sensor). In addition to the stability of the clock frequency and the necessity of falling between the minimum and maximum allowed timing values, there are other time requirements that are essential for successful communication between the driver and the sensor. These requirements consist of the speed of the rising (and falling) edge of the clock signal and the temporal distance between the rise (or falling) of the start pulse with respect to the clock pulse and the digitization speed of the output signal, which must conclude within a duration of approximately half a clock period to avoid an overlap between the acquisition of one pixel and the next. Figure 3 shows, for example, the indication given in the datasheet of the Hamamatsu model regarding the speed of the rising and falling edges and the delay between the clock and the start.

Although very high frequencies are not required, the electronics responsible for driving these sensors must still be able to maintain the high–low transition of the logical states of the digital outputs within the limit of a few tens of nanoseconds. Moreover, at the same time, the stable sending of a train of pulses with a period of a few microseconds and the simultaneous digitization and storage of the input analog signals must be guaranteed. The clock speed of the ATmega processor on almost all Arduino boards is 16 MHz, so the execution time of an elementary operation is in the order of 60 ns. A high-level computer instruction requires numerous elementary operations and, consequently, some periods, or a few dozen periods, of execution time for a single instruction. For more stringent time requirements, it is possible to use the Arduino DUE board which has a microcontroller with a clock that is about five times faster. The language used to program the processor was C/C++, and the high-level instructions that generate a square wave signal, i.e., the ‘clock’ signal, are as follows:digitalWrite(10, HIGH); //set pin 10 to logic level ‘1’;digitalWrite(10, LOW); // set pin 10 to logic level ‘0’.

By executing a cycle in which these instructions are continuously sent to the processor, a square wave signal whose period is equal to 9 μs is obtained, that is, a whopping 150 times the elementary operation duration. Using this signal as an oscillator for the sensors, it is clear that neither the Toshiba chip nor the CMOS camera would be manageable (see Table 1). For the other sensors, this value is within the limits, and the signal transfer of all points shall occur within an acceptable period of time (with a maximum of approximately 37 ms in the case of the Hamamatsu sensor, for which the reading of a pixel occurs every four cycles of clock pulses). It is not difficult, however, to increase the frequency of the square wave emitted by the digital pins of the Arduino, resorting to direct manipulation of the processor registers, rather than using high-level instructions. The two instructions given previously, if transformed into low-level ones, would change as follows:PORTB = 0b00000100; // set pin 10 to logic level ‘1’;PORTB = 0b00000000; // set pin 10 to logic level ‘0’.

This sequence of instructions generates a square wave with a period of only 125 ns, just two elementary cycles. Figure 4 shows the waveforms recorded by an oscilloscope corresponding to a train of pulses sent from an Arduino UNO board using high- and low-level instructions respectively.

If the clock speed appears to be sufficient, we need to verify whether the digitization speed of the sensor output signal sent to an analog input of the Arduino board is also within the limits. This digitization must necessarily be inserted within the clock cycle; therefore, the clock/read cycle should follow this scheme: -Send a high clock signal;-Read and digitize the analog signal output from the sensor;-Send a low clock signal.

In this way, however, since the reading operation lengthens the high-clock time, the pulses no longer have a 50% duty cycle, which is required (even if not strictly) by the sensor electronics. To bring the cycle close to 50%, therefore, the times must necessarily be extended. To find out the digitization time of a signal using high-level instructions, we just run the following loop:for (*n* = 1; *n* < 10,000; *n*++);int value = analogRead(analogPin).

This cycle of ten thousand digitizations is completed in approximately 1.1 s; therefore, a single operation takes place in 100 μs. This is an unacceptable speed, because it should be comparable to at least the slowest clock period (9 μs). Beyond the fact that 100 μs is excessive, even for the Hamamatsu chip (as well as for the Toshiba one), the reading of the 1024 pixels would take place in almost half a second, which would be intolerable if several readings per second were expected. Actually, the digitalization speed is limited by an Arduino hardware setting that can be changed with a few instructions. In practice, the ADC also has its own clock, which is derived from the main clock (the 16 MHz one) scaled by a numerical factor. This factor, called the ‘prescaler’, is equal to 128 by default, which, when combined with the 13 cycles used by the digitization instruction, leads to the 100 μs previously measured. Decreasing the prescaler from 128, for example, to 16, brings the ADC clock to the value of 1 MHz. This is achieved via the following instructions (see Appendix A):#define cbi(sfr, bit) (_SFR_BYTE(sfr) &= ~_BV(bit))#define sbi(sfr, bit) (_SFR_BYTE(sfr) |= _BV(bit))sbi(ADCSRA, ADPS2);cbi(ADCSRA, ADPS1);cbi(ADCSRA, ADPS0).

By setting the prescaler to 16, the time to complete the same cycle seen previously becomes 150 ms, that is, 15 μs for each analog–digital conversion. If the ADC clock is further pushed up to 2 MHz (prescaler equal to 8), a complete cycle that includes sending the clock signal to the sensor, during which the voltage signal is read and digitized, would have a period of approximately 16 μs (8 μs for the conversion and another 8 μs to have a 50% duty cycle). Using this cycle duration, Table 2 shows the times of a complete scan for each of the linear sensors (for the sake of comparison, in the table we also report those that do not allow such a slow clock). For completeness, the same information is shown if the Arduino DUE board is used (optimizing the ADC times). However, if a light sensor does not require a particularly high refresh rate, the board with the 16 MHz processor is suitable for this purpose.

## 5. Results of the Tests Carried out on the Linear Sensors

All sensors presented in Table 2 and shown in Figure 5 were interfaced to an Arduino board to verify the ability of this board to drive the sensors and acquire data as well as to evaluate the reliability of a measurement device.

### 5.1. TSL1401CL

The first sensor to be tested was the AMS model TSL1401CL [21]. This sensor, prepared for surface soldering, was soldered to semirigid wires and put onto a prototype board. For the test, the board was inserted into a light shielding container with a 2 mm wide slit centered above the sensor, 2 cm away. The timing requirements for this array are rather moderate and the number of pixels is low, so we decided to work at a low clock frequency using exclusively high-level instructions. The sequence of pulses sent to the sensor is illustrated in Figure 6. After a while, the program was started, Arduino sends a START signal to empty the content that the photodiodes have previously accumulated. After this signal, the light is collected by the photodiodes until the second START signal. In this sensor, the minimum light integration time (the interval between two START signals) is equal to 110 times the clock duration plus 20 μs. Using high-level instructions, therefore, with a minimum clock time of 9 μs, the minimum integration time is equal to about 1 ms.

After launching the acquisition program, we used two cycles of 150 clock signals plus another cycle of 128 pulses for the voltage reading. In order to adjust the amount of light recorded by the sensor to exploit the maximum signal/noise ratio, a parameter was inserted into the program that allows us to increase or decrease the clock period and therefore modify the integration time. In this way, the time taken for a complete acquisition ranges from a minimum of 16 ms to a maximum of 40 ms. This procedure immediately gave good results.

### 5.2. LF1401

The situation for the IC Haus sensor [22] is similar to that of the AMS. The two chips, in fact, are identical in terms of both the pixel dimensions and electrical characteristics of the electronics. Therefore, the test carried out on this sensor gave results comparable to those of the TSL1401 using the same instructions. The IC Haus chip, however, has some advantages over the AMS one. First of all, the sensor is placed on a plate equipped with holes, on which the electrical contacts can be soldered with a passing-through wire (see Figure 5). Therefore, unlike the twin TSL1401, there is no need to perform surface soldering and the risk of contacts detaching is drastically reduced. Furthermore, in the LF1401, the signal integration time can be directly adjusted using a dedicated input. If this input is connected to the ground, integration is active, while if it is connected to the power supply, integration is inhibited. In this way, the integration time can be adjusted down to zero, which can be useful in case of particularly intense light sources, or increased as desired without having to change the clock period and the acquisition time.

### 5.3. S9226

The Hamamatsu sensor [23] is similar to the previous two in size but contains eight times more pixels. Among all of the sensors taken into consideration in this paper, this is the most expensive and not even the one with the most interesting features, but it is a device with excellent quality and reliability. Unlike 128-pixel devices, this one comes in a package that resembles a common dual in-line package integrated circuit (as the Sony and Toshiba sensors discussed below). In fact, it is equipped with pins that can be inserted into a common eight-pin socket, so there is no need to perform soldering. The number of pixels, although not at the level of the Sony or Toshiba model, is still very high, and the size is the smallest among the linear arrays. A small pixel accomplishes the accuracy of the measurement of the width of a light line, since, for the same-sized illuminated area, the line will intercept a greater number of pixels, thus reducing errors due to background noise or the non-uniformity of the response. This chip requires a maximum value of 100 μs for the clock oscillation period, which is equal to the duration of the digitizing operation on an ATmega board when using high-level instructions (see Section 4). Furthermore, some requirements regarding the rise/fall times of the pulses and the duty cycle (see Figure 3) are such that it is better to drive this chip through the Arduino DUE board. Furthermore, this board works at 3.3 V and can be used to power the S9226. The digitization time of the Arduino DUE is only 4 μs (using high-level instructions), and it is possible to increase the resolution of the conversion up to 12 bit. The electronics of this chip provide data every four clock cycles. Then, the minimum acquisition time of 1024 pixels is just over 32 ms without the need to manipulate the processor registers. As in the case of the 128-pixel arrays, with the Hamamatsu chip, the first tests gave a comforting outcome, so we decided to design and create the new solar compass with this sensor, as detailed in Section 7.

### 5.4. ILX554A

This Sony chip [24] has the only drawback of no longer being in production, although, at the time of writing this paper, many samples are still being sold by several international retailers. Compared to the sensors just illustrated, the ILX554A has the notable advantage of having a much larger number of photodiodes (2048) and a very small-sized single photosensor (14 μm instead of the 63.5 μm of the 128-pixel sensors and comparable with the 7.8 μm of the Hamamatsu one). Moreover, it has a much wider sensitive length (28.7 mm versus approximately 8 mm), and therefore, it is more suitable for precision measurements such as the reading of bar codes or the measurement of electromagnetic spectra. On the other hand, the large number of pixels is not suitable for a modest clock speed, and even storing 2048 values requires hardware that is beyond the capabilities of the Arduino boards. The Arduino analog–digital converter, in fact, works at 10 bits, which implies the use of 2 byte variables (unless you wish to lose resolution or make complicated algorithms to pack 10 bit numbers into 1 byte variables). Therefore, storing 2048 numbers requires a 4096 byte RAM, double that available for the UNO and NANO boards and half that of the MEGA2560 board. A trick that is useful for testing the Arduino-ILX554A combination is to memorize only part of the array or to send the digitized data to an external interface such as a display or a computer connected serially. The latter system allows us to store the data (by the computer), but it has the disadvantage of lengthening the acquisition times, since between one digitization and another, it is necessary to send the data to the serial port. Another possibility is to use the Arduino DUE board, which is equipped with 96 kB of memory and also has a higher clock speed. On the other hand, the 3.3 V working voltage of the Arduino DUE is technically incompatible with that of the Sony chip, which is 5 V. To use this board, therefore, you need a voltage signal adapter. From an electrical point of view, this sensor is slightly more demanding than those previously seen. The clock, in fact, does not set an upper limit to the oscillation period while, on the contrary, the rise and fall times must be contained within 100 ns, and the duty cycle can vary from 40% to 60% at most.

With these assumptions, we decided to verify the possibility of driving this sensor via an Arduino UNO board Rev. 3 using low-level instructions to produce a fast clock and storing the 1500 central pixels using only 1 byte per pixel, thereby decreasing the analog-to-digital resolution to 8 bits. To exploit the full potential of this sensor, a prototype that uses an Arduino to drive the ILX554A could be interfaced to an external memory (an EEPROM or an SD card) to save all 2048 values in a reasonably short time. Alternatively, if the objective of the measurement is to find a single peak, after having verified the presence of a peak (without storing the data), a second scan could be restricted to those pixels that are located around the peak itself. The algorithm for driving and downloading data from the ILX554A sensor does not change much compared to that used for the TSL1401, apart from the use of low-level instructions and with the prescaler of the analog–digital converter decreased from 128 to 32 (see Section 4), providing a frequency of 0.5 MHz. In order to respect the 50% duty cycle, however, the two clock half-periods, where digitization is performed during the first, must have similar durations. With the instructions used, the digitization time was 28 μs, so each half-oscillation of the clock in which there was no digitization was lengthened to 20 μs via a software delay (with a duty cycle of 20/48 = 42%).

The global acquisition time, taking into account a period of 48 μs, an integration time of 20 clock cycles, and 2088 cycles for a complete scan (40 optically inactive pixels and 2048 active pixels), was approximately 100 ms. 

A problem that must be taken into consideration when testing such sensors, which are extremely sensitive to light, is the difficulty of distinguishing a malfunction of the entire sensor from saturation due to excessive light intensity. The photodiodes of ILX554A saturate when hit by a light intensity of 4 millilux for one second. An excessive amount of light compared to this limit, which hits even just a part of the sensor, does not simply lead to a maximum signal (which, for this sensor, corresponds to a minimum output voltage), but to anomalous behavior of all of the pixels, like a broken sensor. To minimize this problem and verify the correct functioning of the chip, we shielded almost the entire sensor with cardboard, leaving it uncovered on a few millimeters, and we inserted the sensor in a box that was completely closed, except for a hole of approximately 6 mm^2^, that was not directly facing the sensor, from which light could enter. In this way, a very small amount of ambient light could reach the sensor area. Figure 7 shows images of the behavior of the signal acquired with the oscilloscope under these conditions, the same signal in which the central pixels are weakly saturated, and finally, the signal obtained when the box containing the sensor was opened on one side. Taking care to avoid sensor saturation, the trial test of the ILX554A via Arduino UNO had a positive outcome.

### 5.5. TCD1304DG

This Toshiba sensor [25] appears to be even more promising than the ILX554A if you need to make highly accurate measurements. Its 3648 pixels of 8 μm width, in fact, if used to measure the angle of a solar ray passing through a slit as in the case of our solar compass, can cover an angular aperture of greater than 70 degrees with a resolution of 2/100 of a degree per pixel. Furthermore, this sensor is still produced by the parent company, and its cost is definitely low. Unfortunately, this was also the only sensor that could not be driven with Arduino. Probably, the request for a very high clock time (minimum frequency 800 kHz) combined with a rather demanding requirement regarding the synchronization between the integration pulse and the start pulse (not exceeding a microsecond) did not allow us to obtain any useful results. In fact, the signal output from the sensor has never provided a trend proportional to the light intensity. Even the use of the Arduino DUE board did not give positive results. The only viable solution at the moment is to have additional electronics to send the clock to the chip and synchronize the other two signals (SH and ICG pins) and to delegate the Arduino only to the digitization of the signal.

## 6. Arduino Interfacing a Matrix Point Sensor

Interfacing a matrix sensor is significantly different from that of a linear array. Matrix sensors (CCD or CMOS) are essentially used for imaging, that is, reproducing a subject through the use of a lens and, possibly, ensuring an image processing speed such that it is attainable to view or record a movie; therefore, it should have no less than 15 frames per second. Unlike the sensors previously seen, in the OV7670 [26], the signal is digitized at the start signal, and the value corresponding to the brightness of the single pixel is sent at each clock cycle via a parallel connection made up of eight pins. At each clock cycle, therefore, the microcontroller must simultaneously read eight digital inputs. Although paralleling eliminates the slow timing problem of serial communication, a camera with VGA resolution (640 × 480 pixels) used to shoot color video at 15 frames per second must send 640 × 480 × 2 × 15 = 9216 million of pixels per second (the sensor uses one byte for chrominance and one byte for luminance information). The required clock speeds, at least theoretically, should not be smaller than 0.1 μs. This requirement is extremely stringent, and a 16 MHz board cannot achieve it. Even if using just one byte per pixel (obtaining a black and white image), the clock would be excessively short. Furthermore, the 300,000 bytes that make up the matrix require a memory that goes beyond that of any Arduino board. This prevents storage for subsequent processing directly on the board’s hardware, and the use of external memory becomes essential.

However, there is a commercial version of the OV7670 camera where, on the same sensor board, there is 384 kB of memory and an oscillator responsible for sending the fast clock to the CMOS. When using this camera, the Arduino processor interfaces only with the memory, whose clock is less demanding (it can reach 1 μs) and also sends commands directly to the sensor and does not need to keep the data for processing, because it can exploit this external memory to read the already acquired frame, even several times. The disadvantage is a small increase in price (of a few euros) and an increase in time, since there is an extra step. On the other hand, the hardware is simplified, and for our purposes, the use of a microsecond clock guarantees an acceptable acquisition time (about 300 ms). The described version is an OV7670 camera with AL422B memory [27] (see Figure 8). For the sake of clarity, we also tried to use an OV7670 without the AL422B, but the result was never very reproducible when we tried to exploit the maximum resolution. The problems essentially arose from electromagnetic noise that interfered with the clock signal (sent by Arduino and traveling along shielded cables for about 30 cm), so that the images obtained from the CMOS were often affected by spurious signals.

The use of the same sensor, equipped with external memory, however, gave much more satisfactory results. Compared to linear sensors, the connections between the board and sensor are much more numerous. As already mentioned, the signal is sent on eight lines in parallel, rather than on just one serial line. There are two communication channels (by I^2^C protocol) for the setting commands and for other signals like the clock and reset. We used the bare minimum number of pins, i.e., 17 out of 22 (see Table 3).

The board used was an Arduino DUE and the instructions both for the clock and for reading the digital pins were written by directly driving the processor registers. 

The sequence of operations to be performed before capturing a picture is quite complex and concerns the setting of the CMOS registers. There are 201 registers and a large number of them must be modified compared to the default value for the entire image acquisition chain to be successful. Fortunately, algorithms can be found on the web that include instructions for setting these registers and that can be adapted according to specific needs. In Appendix B, for the benefit of those who wish to venture into interfacing an Arduino board with the OV7670, there are instructions for the initial setting of the registers to be modified and which, in our case, produced a positive outcome. This CMOS can be set with different resolutions: 120 × 160, 320 × 240, and 640 × 480. Considering that resolution is an important element in any type of measurement and that the acquisition time was not decisive, as long as it was below 1 s, we set the program so that the memory exclusively acquired a VGA resolution image (that is, the highest). Furthermore, since a black and white image was more than sufficient for our purposes, we reduced the signal of each pixel to one byte. Even with only one byte per pixel, however, a 640 × 480 byte matrix exceeds the memory limits of the Arduino DUE. Therefore, to verify the correct acquisition of a VGA frame, there were only two possibilities: dividing the frame into sectors and sending them to a computer for an “a posteriori” entire image reconstruction or giving up storing the bytes in Arduino, sending them directly to the computer. Initially, we checked the correct functioning of the electronics by acquiring a small portion of the entire frame, after which we opted to send the bytes directly to the computer. This solution involved a strong increase in the clock time due to the serial communication timing and, since the AL422B limits the oscillation time of the reading clock (one microsecond at most, according to the datasheet), the insertion of a command to send any input data to the serial port could lead to malfunctions. The duration of the oscillation of a clock pulse, in fact, was found to be equal to 40 μs, of which more than 39.5 μs was due to the use of the serial line (despite using a baud rate of 250,000). Despite this, the test was successful, and it was possible to obtain the VGA image of Figure 9, in black and white, in about 12 s. The sequence of instructions to send to the chip, after setting the parameters, is quite simple and is summarized here in pseudo-code:-Wait for the vertical sync pulse (start of frame);-Send a reset pulse to the WRST pin;-Enable writing to memory by raising the WR pin;-(at this moment, the AL422B memory acquires the image from the camera);-Wait for the vertical sync pulse (end of frame);-Disable writing to memory by lowering the WR pin;-Lower the RRST pin to bring the read pointer to the beginning of the frame;-Wait a few clock cycles and then raise the RRST pin;-Now, at each clock cycle, the byte relating to the brightness of each pixel arrives on pins D0–D7, sequentially.

Without inserting the serial transmission between one clock and another, the acquisition speed of an entire frame, considering the waiting times of vertical synchronization, is approximately 300 milliseconds. Any mathematical processing on the image, therefore, can take place extremely quickly without prejudice due to the fact that it is possible to store only a portion of the entire frame on the Arduino. The presence of the AL422B external memory, however, allows you to acquire a frame and then download various portions of the same image several times in successive moments.

## 7. Design and Construction of a Solar Compass with Arduino and a Linear Sensor

The ENEA Solar Compass is a pointing measurement instrument, patented in 2013 [18], that is able to accurately determine the viewing direction orientation of any observed object of interest with respect to the observation point. The operating principle of a solar compass is based on the knowledge of the position of the Sun in the sky, that is, its angular coordinates viewed from a particular observation point on Earth. These coordinates consist of the elevation of the Sun above the horizon and its azimuth, i.e., the angle between the vertical plane containing the Sun and the local meridian plane (the plane where the two Poles and the observation point lie) [19]. Once the direction of view of the Sun has been measured with respect to a vertical reference plane on the solar compass, we consequently know the angle that the direction of view of the reference plane makes with respect to the geographical north.

Thus, on one side, the instrument is based on a dedicated simplified algorithm for the calculation of the ephemeris and, on the other hand, on the measurement of the direction of the Sun with respect to the vertical reference plane of the compass, obtained with an innovative electro-optical device which uses a slit and an optical sensor. The measurement of the direction of the Sun is performed by calculating the center of the line of light projected on the sensor (2-D or 1-D) with respect to the reference column (2D case) or pixel (1D case). 

This instrument has been proven to be one of the most precise methods for determining true north (with an RMS accuracy within 1/100 of a degree). Unfortunately, the prototype that we built and patented with a 2D sensor cannot be replicated, because some components are no longer available on the market. Anyway, since a linear sensor is sufficient to measure the line of sight of the Sun, after the first tests were carried out on the Hamamatsu chip, we decided to design and create a new solar compass based on this sensor and an Arduino board. Clearly, the accuracy of such an instrument is highly dependent on the geometrical features of the optical configuration. In order to keep the system compact and to avoid excessive broadening of the light line, the sensor cannot be placed at a distance of more than 25 mm from the slit. Then, keeping the sensor at a distance of 20 mm, for example, to distinguish the direction of view of the Sun with a resolution of 0.1 degrees, a sensor with pixel size of less than 35 μm is required. In this respect, the 7.8 μm size of the Hamamatsu array is a very good choice.

The electronics supplied with the compass, in addition to these two elements, include a GPS receiver, an alphanumeric display, an SD card reader, and a Bluetooth module. Everything (except the GPS) is inserted into a console which also contains the battery pack and the voltage regulation modules; externally, there are buttons, LEDs, and various connectors. The linear sensor is placed, as in the original compass, in an aluminum case to be placed above a pointing system, such as a theodolite. The wall facing the Sun, behind which the sensor is placed, is inclined at 45° and presents a vertical slit for the passage of Sun rays. The slit is 4 cm high and approximately 70 μm wide and has been realized by the Institute of Photonics and Nanotechnologies of the CNR in Rome by depositing a thin layer of metal (chromium) on a glass slide and removing part of it using a lithographic technique. The compass head and electronics are connected via a common eight-pole ethernet cable.

Once all the interfaces connected to the Arduino board were verified, the most difficult problem to solve was to calibrate the electro-optical device. In particular, it was necessary to know, with high accuracy, the distance between the sensor and the slit (D), the identification number of the pixel representing the projection of the slit on the sensor (XR0), the two rotation angles of the sensor with respect to the vertical plane containing the slit (α) and the direction of the slit itself (β). Finally, also the deviation angle (ψ) along the horizontal plane between the direction of the vertical reference plane of the compass compared to that pointed to by the sighting system (a telescope) of the support where the compass head is placed (see Figure 10) has to be taken into account. This is because, due to mechanical errors, the last two planes might not coincide.

Calibration took place as follows: the compass was placed on a theodolite equipped with a spirit level and a high-precision goniometer. Then, some LEDs were placed in front of the compass along a vertical line to simulate the Sun at different heights. For each of these LEDs, the pixel number *x_a_*, corresponding to the center of the light line formed on the sensor, was measured for different positions of the goniometer. By plotting the values of this line center as a function of the angle of the goniometer, the result obtained is a series of points placed along different lines, each corresponding to a different height of the light source. 

The slope and *y*-axis intercept of these lines depend on the construction parameters of the compass, so by acting on these parameters, it is possible to optimize the correspondence between the theoretical curve and the experimental data. Assuming that the sensor is mounted perfectly parallel to the wall with the slit and orthogonal to the slit itself (i.e., with α = 0 and β = 90°), the equation that determines the point *x_a_* hit by a ray of light coming from a source placed at elevation φ and azimuth *ϑ* is the following, where *x_a_* and *XR*0 are given in mm (details are in Appendix C):(1)$$x_{a}=D\sqrt{2}\frac{sin(\vartheta )}{\cos\left(\vartheta \right)+tan(\varphi )}+XR0$$

The distance *x_a_* (measured with respect to the edge of the array, i.e., with respect to the first pixel) is exactly *XR*0 when *ϑ* = 0, i.e., when the light is orthogonal to the sensor. The values of *ϑ*, *φ*, and *x_a_* are obtained experimentally, while the *D* and *XR*0 values must be determined so that the difference between the two members of Equation (1) is minimized. In Figure 11, it is possible to appreciate the sensitivity to these parameters by observing the difference between the experimental points and the straight line obtained through Equation (1) for a particular elevation value *φ* and for different values of the parameters D and XR0. 

Once the calibration parameters have been determined (*ψ* is measured successively), the second step is to check the behavior of the compass once it is exposed to the Sun. 

The correctness of the parameters can be estimated by observing the azimuth values provided while keeping the compass stationary for a certain period of time. In this case, its azimuth should be constant since, regardless of the apparent motion of the Sun, the compass always points in the same direction. A trend other than a series of values oscillating around an average value, for example, values that tend to increase, decrease, or form a parabola, proves that some calibration parameters are wrong. In that case, by using specifically developed software to simulate the behavior of the compass, it is possible to change the values of the construction parameters to see the expected output given by the compass and to identify the parameter responsible for that wrong result. 

To this end, we performed several scans with the compass fixed and illuminated by the Sun and recorded a series of line center values *x_a_* vs. time. At the same time, we used our ephemeris calculation software (written in C++ using Microsoft Visual Studio Community 2019 (release 16.11.21), which, by entering the geographical data as the date, time, latitude, and longitude, provides the azimuth of the Sun) and calculated the foreseen position of the light line on the sensor, depending on the given calibration parameters. 

In this way, firstly, the correctness of the calibration parameters is verified by comparing the sequence of theoretical and experimental azimuth values and, secondly, if necessary, the estimate of the parameters can be refined by forcing this trend to be constant (with the sensor stationary, as already mentioned, the azimuth of the compass should not change). The simulation program, moreover, is able to optimize the parameters automatically, leaving the possibility of changing them manually. Figure 12 shows both the azimuth measured by the compass and the azimuth resulting from the simulation with this program (left graph) and the result after optimization (right graph). 

Once *D* and *XR*0 have been optimized, the deviation between the reference plane of the compass and the direction aimed at by the pointing system (the angle ψ in Figure 10) remains to be included in the calculations.

In order to estimate this last parameter, it is crucial to know the true azimuth of a given target seen by the observation point. If the azimuth between points A (observer) and B (target) is known, in fact, it is sufficient to aim at point B from A, measure the azimuth provided by the compass, and calculate the difference between this value and the known azimuth. To obtain, with a suitable level of accuracy, the true azimuth of the line joining points A and B, it is necessary to know the geographical coordinates of the two points and use a method that takes into account the curvature of the Earth [28]. For a typical accuracy of a few meters on the geographical coordinates, a distance between A and B of larger than 10 km is sufficient to reduce the error on the true azimuth to values smaller than the error on the experimental azimuth given by the compass. Obviously, the ψ angle should be measured any time the compass is removed from the pointing instrument and then placed again on it, unless a very accurate mechanical system for a reproducible positioning is adopted.

Once this last calibration phase has been completed, the compass is ready to be used as a measuring instrument. 

A few days after completing the whole calibration procedure, we checked if the compass continued to provide the correct azimuth values. Therefore, we placed the compass on an observation point at the ENEA Center in Frascati, whose azimuth values with respect to some reference points, located in Rome, are known (using the geographical coordinates method, as described above), and we carried out some measurements. The results are shown in Table 4.

Even if the error of a single measure doubles when compared with the 2D-sensor case [19], these data show that this device has an error of less 0.5 arc minutes (0.008 degrees) during repeated measurements.

Although the solar compass is a very old orientation tool, with some interesting handy examples [29], it is still one of the most accurate devices for determining the geographic north. Apart from magnetic compasses, which point towards the magnetic pole and not towards the true geographic north, other kinds of compasses are based on the gyroscopic effect and on GPS satellites.

Gyroscopic compasses are the most accurate ones (with an uncertainty of a few arc seconds), but their cost is very high (tens of thousands of euros), and the time necessary to build and align the setup is very long [30]. The compasses based on GPS satellite constellations are cheaper than gyroscopic compasses, but their accuracy is worse than our electronic solar compass. Recent works testing the validity of GPS measurements provide accuracies ranging from 0.07 to 1.5 degrees [31] and from 0.09 to 0.21 degrees [32], and, in a paper concerning a measurement campaign for the orientation of paleomagnetic drill cores, the RMS deviation of a GPS compass was shown to be well above 0.1° [33].

Therefore, with respect to other survey devices, we can conclude that our solar compass, based on the Arduino–Hamamatsu pair, is an instrument with excellent accuracy and compactness and a competitive cost.

## 8. Conclusions

The versatility and reliability of Arduino boards are well known, and the possibility of finding a considerable quantity of sensors on the market at a low price as well as retrieving, on the Internet, the codes for their interfacing, makes them devices that can also be used in a research laboratory. In this paper, we illustrated tests of light sensors, in both linear and matrix arrangements, connected to Arduino boards to verify their compatibility and the possibility of using these systems as measuring instruments. We took into consideration five linear sensors manufactured by AMS, IC Haus, Hamamatsu, Sony, and Toshiba, consisting of a minimum of 128 pixels to a maximum of 3648 pixels and a 640 × 480 pixel CMOS camera from Omnivision. In all tests, except for the Toshiba sensor, the trials were positive, and we were able to verify how the Arduino/photosensor combination could be a valid device for measurements involving light, such as a spectrometer or a reader of bar code, or even for imaging systems. In particular, a solar compass was designed and created based on an Arduino DUE board and the Hamamatsu chip for measuring the direction of geographic north (or the viewing direction of any object of interest with respect to geographic north) by exploiting the equations of terrestrial motion. This solar compass obtained results that are similar to those achieved with the patented ENEA solar compass, whose performance is, in turn, comparable with the best devices available on the market, but whose cost is at least two orders of magnitude greater than that of our device.

## Figures and Tables

**Figure 1 sensors-23-09787-f001:**
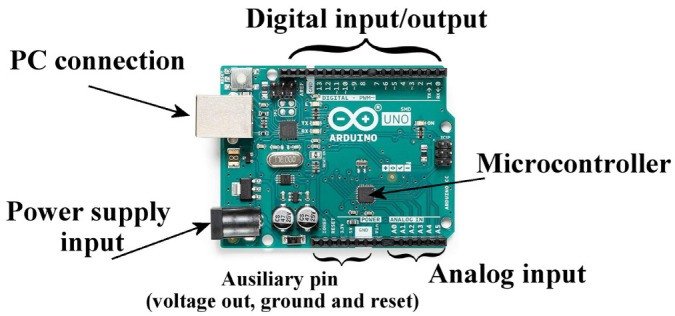
The Arduino UNO board with its main components.

**Figure 2 sensors-23-09787-f002:**
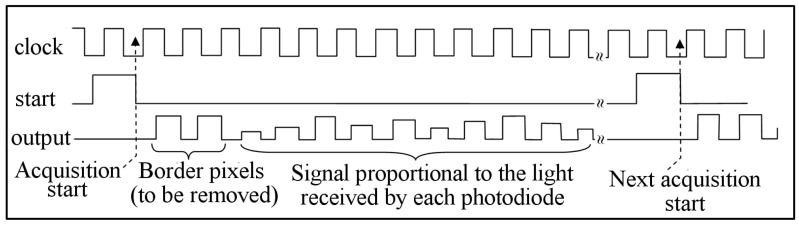
Example of the sequence of the clock, start, and output pulses for the acquisition of the light signal.

**Figure 3 sensors-23-09787-f003:**
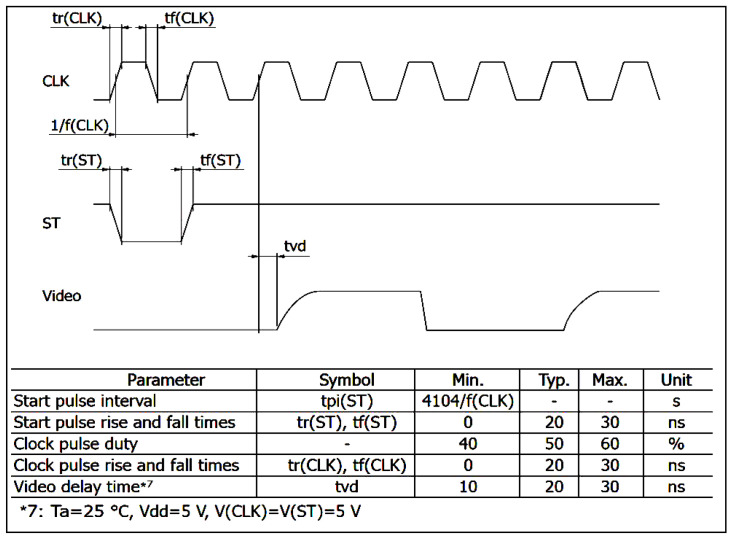
Rising and falling edge requirements of the Hamamatsu sensor.

**Figure 4 sensors-23-09787-f004:**
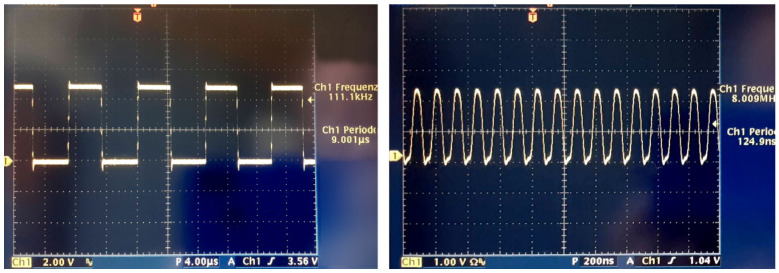
A square wave emitted from an Arduino UNO pin at the maximum frequency. (**Left**) The output signal when high-level instructions are used (the sharp peaks visible at any voltage flip are due to imperfect balance of the electronic circuit); (**Right**) the same signal when the processor registers are directly manipulated. Note that the time scale of the right figure is 20 times shorter than the left one.

**Figure 5 sensors-23-09787-f005:**
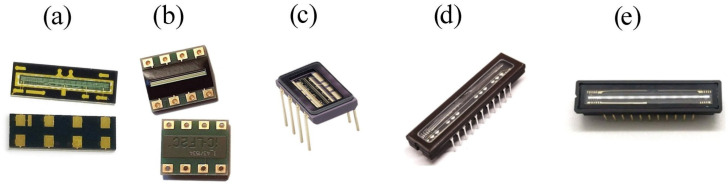
Photos of the five sensors interfaced to the Arduino boards: (**a**) TSL1401CL, (**b**) LF1401, (**c**) S9926, (**d**) ILX554A, (**e**) TCD1304D.

**Figure 6 sensors-23-09787-f006:**
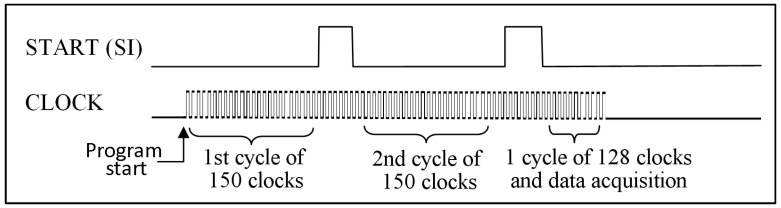
Sequence of the pulses sent by Arduino to the TSL1401CL sensor for the acquisition of the 128 pixel signals.

**Figure 7 sensors-23-09787-f007:**
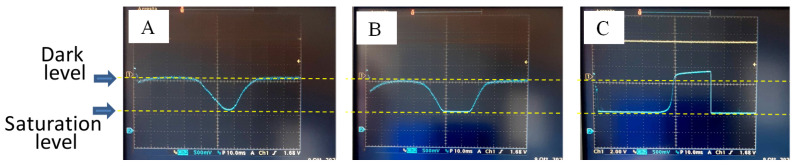
Light signals measured by the ILX554A sensor, driven by an Arduino UNO and recorded by the oscilloscope. (**A**) Signal under normal conditions (curve that drops to a minimum); (**B**) signal under weak saturation conditions; (**C**) signal under strong saturation conditions. Situation (**C**) could be mistaken for a malfunction of the sensor, since it seems to saturate in an unlit area and give zero signal (indeed, even below the minimum) in the part hit by the light.

**Figure 8 sensors-23-09787-f008:**
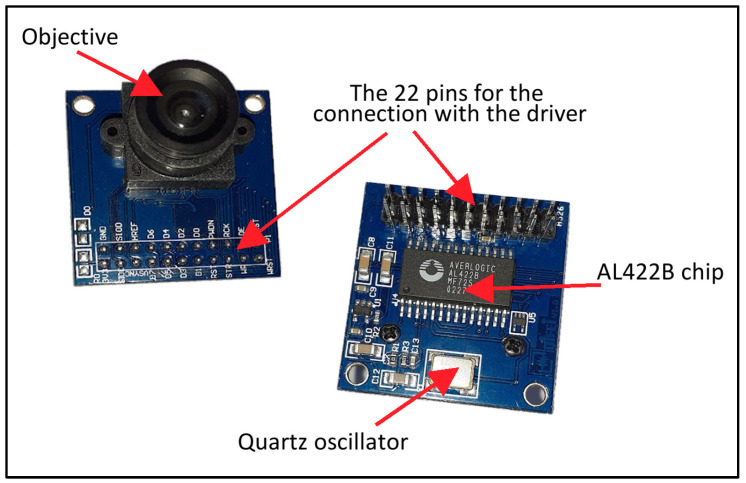
The camera OV7670 with the AL422B memory chip. The classic version does not have the integrated circuit on the back side, and there are 18 connection pins instead of 22.

**Figure 9 sensors-23-09787-f009:**
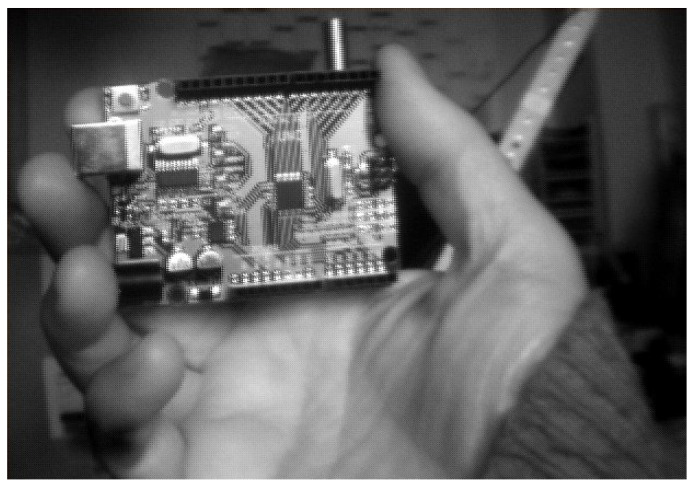
A 640 × 480 pixel photo taken with the OV7670 camera, supported by AL422B memory, interfaced with Arduino.

**Figure 10 sensors-23-09787-f010:**
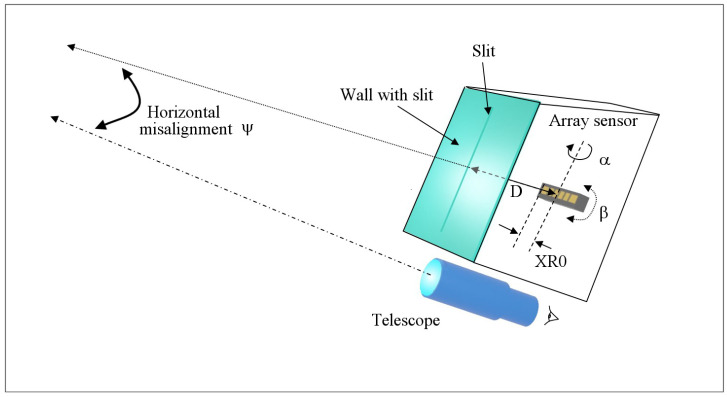
Representation of the variables and parameters to take into account when calibrating the solar compass.

**Figure 11 sensors-23-09787-f011:**
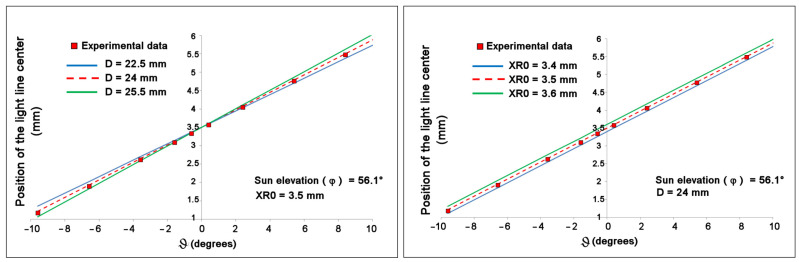
Line of light position measured on the array, from a light source with an elevation of 56.1° and its azimuth varied with respect to the normal of the sensor, compared with the corresponding theoretical value for different distances *D* (**left**) and *XR*0 (**right**). We can see how very small differences in parameters easily lead to a disagreement with the experimental data.

**Figure 12 sensors-23-09787-f012:**
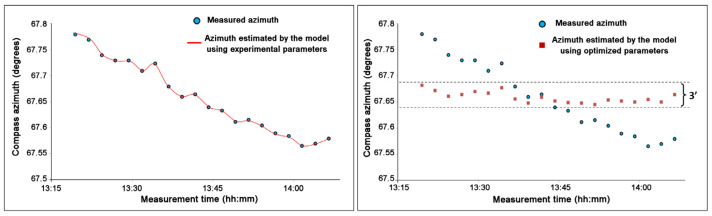
(**Left**) Experimental data of the azimuth measured by the stationary compass (circles) and the corresponding values calculated by the software (continuous curve), in which the construction parameters of the compass and the light line position, resulting from the measurement, were entered; (**Right**) the experimental data values (circles) and the new data calculated using the optimized parameters (squared). With optimization, by changing only parameter D of 0.4 mm, the points are arranged around a constant value within ±1.5 arc minutes, i.e., ±1/25 of a degree.

**Table 1 sensors-23-09787-t001:** Optical/electrical features and costs of the tested light sensors.

Model	Number of Pixels	Pixel Spacing (μm)	Active Area Length (mm)	Minimum External Clock (μs)	Maximum External Clock (μs)	Estimated Price (€)
TSL1401CL (AMS-Premstaetten Austria)	128	63.5	8.13	0.125	200	10
LF1401 (IC Haus—Bodenheim, Germany)	128	63.5	8.13	0.2	No limits	13
S9226 (Hamamatsu—Hamamatsu City, Japan)	1024	7.8	8	1.25	100	80
ILX554A (Sony—Tokyo, Japan) *	2048	14	28.7	0.5	No limits	10
TCD1304DG (Toshiba—Tokyo, Japan)	3648	8	29.2	0.42	1.25	30
OV7670 (Omnivision—Santa Clara, CA, USA)	640 × 480	3.7 × 3.7	2.36 × 1.76	0.02 **	1 **	15

* This chip is discontinued but can still be found online from some electronics vendors with a very wide price range. ** These refer to the clock of the AL422B chip, an external memory used as a buffer for data. The minimum clock time for CMOS is still 0.02 μs, but the maximum value is only 0.1 μs.

**Table 2 sensors-23-09787-t002:** Acquisition times for a complete scan of the arrays to be tested using a clock of 16 μs for the UNO board and the lowest possible clock (around 3 μs) for the DUE board.

Model	Number of Pixels	Clock Cycles for a Single Pixel Readout	Arduino UNO Acquisition Time (ms)	Arduino UNO Acquisition Rate (Hz)	Arduino DUE Acquisition Time (ms)	Arduino DUE Acquisition Rate (Hz)
S9226	1024	4	66	15	12.3	81.4
TSL1401CL	128	1	2	500	0.4	2500
LF1401	128	1	2	500	0.4	2500
ILX554A	2048	1	33	30	6.2	162.8
TCD1304DG	3648	4	233	4	87.6	11.4

**Table 3 sensors-23-09787-t003:** List of pins on the board with the CMOS OV7670 and the AL422B memory with the indications of those actually used and their functions.

PIN	3V3	SIOC	VSYNC	D7	D5	D3	D1	RST	STR	WR	WRST
Use	Voltage supply	Clock I^2^C	Vertical syncro	Bit 7	Bit 5	Bit 3	Bit 1	→3.3 V	Not used	Write enable	Write reset

PIN	GND	SIOD	HREF	D6	D4	D2	D0	PWDN	RCK	OE	RRST
Use	Ground	Data I^2^C	Not used	Bit 6	Bit 4	Bit 2	Bit 0	Not used	Byte readout clock	→GND	Read reset

**Table 4 sensors-23-09787-t004:** Azimuth values measured using the compass made with the Arduino DUE board and the Hamamatsu S9226 sensor a few days after completing the calibration process and a comparison with the actual values. From these data, both the precision and accuracy of this instrument can be appreciated. Note that the differences between the data obtained by the maps (the expected value) and the experimental data are expressed in arc minutes.

	Saint Peter’s Dome	Broadcast Antenna on Monte Mario	Bell Tower of the Basilica of Santa Maria Maggiore
Number of measures	5	5	3
Measured azimuth (degree)	116.6956°	120.4162°	120.894°
Expected value (degree)	116.6891°	120.4103°	120.8946°
Difference (arc minute)	0.39′	0.35′	−0.04′
Maximum deviation of all measurements (arc minutes)	0.46′	0.43′	0.19′
Standard deviation (arc minutes)	0.37′	0.29′	0.17′

## Data Availability

Experimental data are reported in an internal laboratory notebook and are available upon request to the corresponding author.

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
