# Peer review of "Interfacing Arduino Boards with Optical Sensor Arrays: Overview and Realization of an Accurate Solar Compass"

_sensors, 2023, doi:10.3390/s23249787_

Round 1
Reviewer 1 Report
Comments and Suggestions for Authors
For this work, the team of authors designed a solar compass through Arduino and sensors. Experimental results proved that this solar compass gave similar results to the patented ENEA solar compass, but at two orders of magnitude less cost.
Overall, I think this work is still innovative. Seeing the Arduino development board brought back memories of my undergraduate days. My undergraduate final project was also an Arduino UNO3 based sensor. However, as far as I can remember, it seems like Microsoft has stopped updating this series of Arduino.
In fact, in addition to Arduino, the author's team can also try STM89C52.Besides that, FPGA should also be able to achieve the same effect at a faster speed, but the cost is definitely higher.
The author lays out the details up front. The work is presented only at the end. I don't have much comment on the work in this paper. It should be noted that the formulas on which the author developed the compass based do not seem to be cited, so if there is a source, please ask the author to add it.
Reviewer 2 Report
Comments and Suggestions for Authors
This manuscript entitled “Interfacing Arduino boards with optical sensor arrays: over-view and realization of an accurate solar compass” present the arranged linear arrays of light multi-sensors, which can digitize and store voltage signals precisely and quickly. The paper has been well-prepared and the length is suitable, thus it is suggested to publish this manuscript in a minor revision:
1. Authors have provided the main components of this Arduino UNO board in Figure 1, it is suggested to point out which component is designed by authors and which component is purchased from the company. It is much more meaningful to develop the system based on the purchased components. Please discuss more in this part.
2. In figure 4, there are some sharp peaks for each period, please clarify this part and discuss the underlying mechanism.
3. In the process of the calibrating the solar compass, how about the effect of pixel numbers? Also, how about the effect of the position of the optical sensor arrays? Please describe this part and discuss more.
4. Some important progress towards light sensors are suggested to be included in the introduction part, such as Appl. Phys. Lett. 123, 013507 (2023), Nat Rev Mater 8, 587–603 (2023), Journal of Inorganic Materials, 2023, 38(9): 1055-1061, Materials 2022, 15(23), 8280, J Infrared Milli Terahz Waves 38, 143–154 (2017).
Comments on the Quality of English LanguageMinor revision is suggested.
Reviewer 3 Report
Comments and Suggestions for Authors
Interfacing Arduino boards with optical sensor arrays: overviews and realization of an accurate solar compass have been discussed in the paper. A description of the interface between Arduino and light multi-sensors is presented in this paper to emphasize the potentiality of Arduino boards. There are tens, hundreds, or even thousands of elements in these sensors, and their sizes range from a few microns to tens of microns and can be arranged in linear arrays or in matrix configurations (CCDs or CMOS cameras). Before publication, there are a few minor concerns:
1. There is a need to improve the English.
2. Figure quality is poor, even the captions in the figure are unclear, especially in Figures 11 and 12.
3. In the discussion part, they should discuss and compare a clear literature review.
Comments on the Quality of English Language
Need to improve a lot
